Perspectives

# Beyond Mendel: a call to revisit the genotype–phenotype map through new experimental paradigms

Diethard Tautz [1,2,*,†] Luisa F. Pallares [3,†] Leif Andersson [4,‡] Neda Barghi [1,‡] Nick Barton [5,‡]
Rachael Bay [6,‡] Yingguang Frank Chan [7,‡] Angela Hancock [8,‡] Tobias S. Kaiser [1,‡] Daniel Koenig [9,‡]
Zacharias Kontarakis [10,‡] Miriam Liedvogel [11,‡] Juliette de Meaux [12,‡] Magnus Nordborg [13,‡]
Abraham A Palmer [14,‡] Michael Purugganan [15,‡] Christian Schlötterer [16,‡] Karl Schmid [17,‡]
Didier Y.R. Stainier [18,‡] Detlef Weigel [19,‡] Jochen B.W. Wolf [20,21,‡] Dieter Ebert [2,22,†] Greg Gibson [23,*,†]

[1]Max-Planck Institute for Evolutionary Biology, Plön 24306, Germany
[2]Stellenbosch Institute for Advanced Studies (STIAS), 10 Marais Road, Stellenbosch 7600, South Africa
[3]Friedrich Miescher Laboratory of the Max Planck Society, Tübingen 72076, Germany
[4]Department of Medical Biochemistry and Microbiology, Uppsala University, Uppsala 75123, Sweden
[5]Institute of Science and Technology Austria, Evolution & Ecology, Klosterneuburg 3400, Austria
[6]Department of Evolution and Ecology, University of California Davis, Davis, CA 95616, United States
[7]Groningen Institute for Evolutionary Life Sciences (GELIFES), University of Groningen, Groningen 9747AG, Netherlands
[8]Department of Botany and Plant Pathology, Purdue University, West Lafayette, IN 40797, United States
[9]Department of Botany and Plant Sciences and Institute for Integrative Genome Sciences, University of California, Riverside, CA 92507, United States
[10]Genome Engineering and Measurement Laboratory (GEML), ETH Zürich, Zürich 8093, Switzerland
[11]Institute of Avian Research, An der Vogelwarte 21, Wilhelmshaven 26386, Germany
[12]University of Cologne, Plant Molecular Ecology, Institute of Plant Sciences, Cologne 50674, Germany
[13]Austria Academy of Sciences, Vienna BioCenter, Gregor Mendel Institute, Vienna 1030, Austria
[14]Department of Psychiatry and Institute for Genomic Medicine, University of California San Diego, La Jolla, CA 92093, United States
[15]Center for Genomics and Systems Biology, New York University, New York, NY 10003, United States
[16]Institut für Populationsgenetik, Vetmeduni Vienna, Vienna 1210, Austria
[17]Institute of Plant Breeding, Seed Science and Population Genetics, University of Hohenheim, Stuttgart 70599, Germany
[18]Department of Developmental Genetics, Max Planck Institute for Heart and Lung Research, Bad Nauheim 61231, Germany
[19]Department of Molecular Biology, Max Planck Institute for Biology Tübingen, Tübingen 72076, Germany
[20]LMU München, Biozentrum Martinsried, Department of Evolutionary Biology, Großhaderner Straße 2, Martinsried 82152, Germany
[21]MPI Biologische Intelligenz, Department of Microevolution and Biodiversity, Eberhard-Gwinner-Straße, Seewiesen 82319, Germany
[22]Department of Environmental Sciences, Zoology, University of Basel, Vesalgasse 1, Basel CH-4051, Switzerland
[23]Center for Integrative Genomics and School of Biological Sciences, Georgia Institute of Technology, Atlanta, GA 30302, United States

*Corresponding authors: Diethard Tautz, Max-Planck Institute for Evolutionary Biology, Plön 24306, Germany. Email: tautz@evolbio.mpg.de; Greg Gibson, Center for Integrative Genomics and School of Biological Sciences, Georgia Institute of Technology, Atlanta, GA 30302, United States. Email: greg.gibson@biology.gatech.edu
†These authors are part of the writing group.
‡These authors are part of the supporting group listed in alphabetical order.

The long-standing notion that genotypes map to phenotypes through simple one gene–one trait relationships continues to shape both research in the life sciences and public understanding, with implications for policy and funding priorities. Yet this paradigm is increasingly recognized as inadequate for explaining continuous phenotypic variation and the complex genetic architectures of the genotype–phenotype map. Modern genetics emerged from the early 20th-century synthesis of Mendelian and biometric schools of heredity, with R.A. Fisher demonstrating early on how multiple discrete loci could collectively produce continuous variation. Despite this fundamental insight, Mendelism—with its focus on single genes and standardized genetic backgrounds—became the dominant framework, shaping current genetics research and molecular biology as well as science education. The advent of large-scale genomic data has revealed yet again the limitations of this reductionist approach. Evidence from quantitative genetics now shows that most phenotypes arise from complex networks of many interdependent genes and their dynamic responses to environmental perturbations. Here we trace the historical roots of how Mendelian classical genetics departed from the biometric school to create the current predominant paradigm in genetics, despite fundamentally unresolved issues. Moving on from this one-sided paradigm will require systematic development of integrative, evolutionarily grounded experimental approaches that better capture the multigenic and context-dependent nature of inheritance. Achieving such an extended perspective will require methodological innovation, including advances in large-scale (e.g. automated) phenotyping. Dedicated research programs will be necessary to advance a new era of genetic research into the complex mechanisms underlying phenotypic variation.

Keywords: classic genetics; quantitative genetics; genotype–phenotype map

## The establishment of the dominant view that genes of large effect are the key to understanding phenotypic variation

### The birth of genetics

Two of the major founders of genetics, Gregor Mendel and Francis Galton, were born in 1822. Both generated data and developed concepts for the mechanisms of heredity but with fundamentally different conclusions. Galton was actually much better known than Mendel at the time when he published his fourth book "Natural Inheritance" (Galton 1889), in which he focused on the inheritance of continuous variation, such as human body height. He started with systematic data collection from families and developed the general statistical tools for quantitative genetics that we still use today. This work flourished, especially with further mathematical and statistical elaborations by Karl Pearson in a series of papers called "Mathematical contributions to the theory of evolution" around the turn of the century (starting with (Pearson 1900)), but this "biometric school" of genetics was contested by the rediscoverers of Mendel´s work, among them Hugo de Vries. They emphasized the discrete ("particulate") nature of genetics, as revealed in the distinct allelic states of Mendelian inheritance, and this approach largely displaced the concepts of continuous variation.

The elements of genetics were called "genes" by the Danish geneticist Wilhelm Johannsen, taking recourse to the "pangenesis" concept of Darwin. Johannsen laid down in his textbook "Elemente der exakten Erblichkeitslehre" (*Elements of the Exact Theory of Heredity*) (Johannsen 1909) the full agenda for much of today´s research in genetics. In addition to the term "gene," he also introduced "genotype" and "phenotype." Most importantly, he advocated the use of inbred lines ("Reine Linien") in order to control experimentally for the generally observable variance among individuals. In fact, Mendel had already used such a preselection of lines for his famous pea breeding experiments (Mendel 1866). Johannsen claimed that only the use of inbred lines would reveal the discrete nature of the genes and their effects. He contrasted this specifically to Galton´s teaching of continuous variation.

> *The old controversy as to whether in living nature there are only continuous transitions or real discontinuity must, in our opinion, be decided in favor of discontinuity: There are always continuous transitions between individuals and also between phenotypes—but the real genotypic differences are discontinuous.* ((Johannsen 1909)—page 327; original text is in German)

Another one of Johannsen´s key conclusions was that any character for which a gene can be identified should be a single trait. This new conceptual thinking quickly became the mainstream agenda of genetic research, especially since it offered so many direct experimental options. It inspired Thomas Hunt Morgan both to develop *Drosophila* as a model system and to use dedicated mutagenesis experiments for identifying genes that generate a phenotypic trait. Morgan discovered that the hereditary material is organized into chromosomes, and he published the first genetic maps of chromosomes based on distinct genetic loci in his landmark book *The Mechanism of Mendelian Heredity* (Morgan et al. 1915). Subsequent work along the lines of this new conceptual thinking was honored by many Nobel prizes in the following decades, while the concepts of the "biometric school" disappeared into the background (Morgan did not even mention it in his book).

### Genetics without genes

While experimentalists focusing on discrete phenotypic variation and single genes started to dominate the new field of genetics, the "biometric school" continued to consolidate the statistical tools to study continuous trait variation. First and foremost was the work of Ronald Aylmer Fisher who reconciled the particulate nature of Mendelian loci with the continuous phenotypic variation of the biometricians (Fisher 1918). Fisher's infinitesimal model has been foundational to quantitative trait genetics and became ever more relevant over time (Barton et al. 2023, 2017; Bulmer 1972; Turelli 2017).

The success of decades of such statistical approaches to the genotype–phenotype map was reflected in the consolidation of the field of quantitative genetics (Falconer and Mackay 1996). The development of the new breeds during the agricultural transformations of the "Green Revolution" beginning in the 1960s is the best testimony of its practical importance. By now it is well established that the best predictor of breeding success toward a desired phenotype in plants and animals is not typing just a single or a few marker genes but generating a marker profile across the whole genome, a procedure called "genomic selection" (Meuwissen et al. 2001). This realization supported empirically the fact that traits have a complex polygenic basis and that the response to selection can indeed be predicted by quantitative genetic models that assume very many and very small genetic effects. The success of breeding by genomic selection in the past two decades has completely reshaped agricultural production (Alemu et al. 2024; Meuwissen et al. 2016), but even before we were able to trace the genomic marker profiles, the notion of a polygenic basis of traits was experimentally demonstrated by artificially selecting traits in genetic model systems, e.g. *Drosophila* bristle number (Mackay et al. 1994; Nuzhdin et al. 1995) or mouse body size (Keightley 1998) showing that polygenic variation is sufficient to shape phenotypes in almost any direction.

In spite of the success of this research approach in both, producing a better understanding of the complex genetic basis underlying trait variation and in making practical use of this knowledge, it cannot yet generate a mechanistic understanding of single-gene effects—which for many biologists is a sine qua non for true understanding. This realization was clearly stated by Nick Barton in 2022 (Barton 2022):

> *Paradoxically, natural selection may be most effective in cases where it is least accessible to investigation. If adaptation is polygenic, and function is diffuse, we require a quantitative approach … even if it cannot attempt to elucidate the effects of each distinct Mendelian variant.*

Therefore, one can easily appreciate why classical genetic approaches that aim for a mechanistic understanding of single-gene effects not only became the dominant strategy for experimental geneticists but still govern investments in genetic and molecular biology research.

### The dominant view

Many attempts have already been made to highlight the need for embracing the genetic complexity underlying trait variation, starting from the *Modern Synthesis* in the 1930s and 1940s to more recent post-genome-wide association studies (GWAS) publications (e.g. Barton 2022; Boyle et al. 2017; Rockman 2012; Travisano and Shaw 2013). Yet, these attempts have failed to convince many molecular biologists who remain resistant to the idea that polygenicity can provide meaningful mechanistic insights. The assumption that any experiment that would require

statistical analysis has just been poorly designed remains all too prevalent. Most of the investment into genetics and molecular biology still focuses on the analysis of single-gene effects, increasingly in single-cell or organoid studies and paired with structure analysis of the corresponding molecules. Whole research centers with large infrastructure are set up according to these experimental approaches, usually with the claim that they will eventually allow to tackle all human genetic diseases while largely ignoring the limitations of the classical approach.

In this perspective, we use the term "classical genetics" to refer to the dominant experimental and conceptual view of genetics while acknowledging that there is a spectrum of opinions on what the term refers to. This classical view and the experimental approaches associated with it assume that 1) a gene is defined by its mutant phenotype, that 2) gold standard proof that the correct gene has been found requires reversion of the phenotypic defect by introducing a wild-type copy of the gene into the mutant strain, and that 3) functional analysis of the encoded molecules, be they RNA or protein, is what generates a mechanistic understanding. Given these constraints, it is not surprising that molecular biology became reduced to a small set of model organisms where such complex genetic manipulations are possible.

Indisputably, the success of this agenda has been stellar. It has given us amazing insights into the biochemistry of life and the interaction of cells. This can be righteously considered as one of the most revolutionary achievements of biology in the past decades. However, the insights derived from this approach have still a number of severe limitations that are further outlined below.

## What are the limitations of the classical genetic approach?

Despite the great success of the current molecular genetic experimental paradigm in shedding light on the genes and pathways that are fundamentally important to "make or break" a trait or an organismal function, it falls short when it comes to understanding phenotypic variation across individuals and how a given genome generates a given phenotype. Bridging this divide requires engaging with fundamental genetic phenomena that are well-known but which cannot be understood by studying one gene at a time. In the following sections, we briefly list essentially unresolved genetic issues that serve as reminders of the associated problems, rather than representing extensive treatments of these issues. We conclude that, in their sum, they should send us back to the drawing board for conceptualizing the way in which genetic information maps onto phenotypic variation, a task that requires the development of new experimental and theoretical paradigms and approaches.

### The environmental influence issue

Genotype by environment interactions (GxE) (Saltz et al. 2018; Via and Lande 1985) are rarely addressed by the classical genetics approach, which reports the effect of a gene on a phenotype in an artificially controlled environment, without discussing the potential dependency of such an effect on the specific conditions under which the experiments were performed.

The phenotypic variation within completely inbred strains is collectively ascribed to environmental influences, which is an amalgam of external factors, epigenetic variation including transgenerational effects, and stochastic variability in developmental processes as well as measurement error. This is particularly relevant when considering the reproducibility of the developmental progression that leads to a specific phenotype. The trajectory from a fertilized egg toward an adult organism passes through a series of phenotypic stages and developmental decisions, each of which are subject to environmental disturbance. Waddington was the first to point out that this requires the evolution of genetic robustness, which he called canalization (Waddington 1957). This has become an active field of research, including the identification of genetic systems that confer developmental robustness (Felix and Barkoulas 2015; Tawfeeq et al. 2024) and the integration of the concepts into understanding human genetic diseases (Gibson and Lacek 2020) and polygenic risk scores (Jayasinghe et al. 2025; Xiang et al. 2024). Yet, this entanglement of genetic systems with environmental influences is difficult to reconcile with single-gene activities. We still know very little about which genetic processes contribute to buffering mechanisms and how individual genotypes respond to environmental stimuli (Rossi and Kontarakis 2022).

## The issue of the lack of mutant phenotypes

A classical genetics approach assumes that when a gene is mutated, a clear phenotypic effect should be produced in the trait regulated by the gene. Yet when systematically targeted knockout studies of genes became possible, one of the first fundamental findings was that substantial numbers of targeted genes did not show an obvious phenotype when mutated and did not seem to be necessary for the well-being of the cell or the organism—at least not in a lab environment (El-Brolosy and Stainier 2017). This has often been explained away by saying that it is impossible to test the function of a gene under all of the ecological circumstances that an organism would encounter, and indeed, as more ecologically relevant conditions are tested, more gene functions are usually revealed. This has led to a concept of "essential genes," to be distinguished from "less important" genes, but the concept of "essential genes" is actually itself very fluid. The functions of seemingly essential genes have turned out to be context dependent and quickly evolvable (Rancati et al. 2018). The relative conservation of a gene alone is not a strong predictor of its phenotypic effect in knockouts.

Another possible explanation for not finding phenotypes for given knockouts is genetic redundancy, through either duplicated genes or backup pathways, which can be modeled within the framework of classical genetics (Nowak et al. 1997). But new results, especially from experimental evolution experiments, suggest that some redundancy observations are better explained by a polygenic framework (Láruson et al. 2020; Schlötterer 2023). Hence, understanding why knockouts of many genes do not lead to clear phenotypes remains a largely unresolved challenge.

## The genetic suppressor issue

Suppressor mutations are well-known in classical genetics and have often served to reveal interactions within gene networks. Such gene–gene interactions are generally classified under the term epistasis (Mackay and Anholt 2024) and can be studied to determine the hierarchy of gene actions in networks. However, recent evidence shows that such suppressor mutations do not necessarily reflect static networks but are readily evolvable, which makes epistasis and the effect of a gene on a trait something transient instead of a fixed property as assumed by classical genetics approaches.

Many genes, when mutated, actually do cause a phenotype. Unless the mutation is lethal, one can keep growing the strains carrying the mutation. Would such strains eventually recover from the loss of the gene and become healthy again? Several groups have now systematically explored this question with

stunning results. In yeast, about two-thirds of 180 genotypes with measurable knockout phenotypes reached near wild-type fitness through accumulation of adaptive mutations elsewhere in the genome (Szamecz et al. 2014). Another study in yeast found that losing highly connected genes increased the evolutionary potential by facilitating the emergence of a more diverse array of phenotypes, some even fitter than the original cells (Helsen et al. 2020), and even new cellular morphologies and growth characteristics can evolve in yeast cells as a by-product of such compensatory evolution (Farkas et al. 2022). In *E. coli*, the effects of mutations in fundamental metabolic genes can be rescued in laboratory evolution experiments, resulting in the rewiring of existing hardwired networks (McCloskey et al. 2018). A similar phenomenon—called "transcriptional adaptation," becomes increasingly evident in the context of knockout experiments in medically motivated studies, blurring the concepts of clear genotype–phenotype relationships (Jakutis and Stainier 2021). While second-site suppressor screens have been highly successful in model organisms like *Drosophila*, it is generally also possible to modify phenotypes simply by backcrossing mutations into different wild-type backgrounds (Gibson and Dworkin 2004).

The ease with which suppressor mutations can be generated in nonrelated gene networks remains surprising. One could argue that there might be previously evolved backup networks in the cell that can deal with losses of gene activity. However, it is not clear how these networks are maintained over time if they are not tested regularly in natural populations. Again, although these suppressor phenomena have been well-known for a long time, they remain poorly understood.

## The pleiotropy issue

The idealization that a trait should be regulated by one gene or that the main function of a gene is linked to just one trait is certainly the exception rather than the rule. However, this assumption is still entrenched in classical genetic approaches. While the notion of genes as beads-on-a-string is no longer prominent, the notion that genes generally map singularly to a phenotype is arguably how gene effects are presented to the public.

It is thus noteworthy that when one searches for the function of a given gene based on publications on the gene, one will almost always find multiple answers, where researchers have studied the same gene under different experimental conditions, in the context of different cell types or developmental stages or within different regulatory networks. In fact, most genes, in particular those encoding transcription factors and signaling pathways, are active in multiple developmental stages and tissues, allowing them to perform many functions. In mice, only the most recently evolved genes tend to be restricted in their expression to one tissue, but they quickly add additional expression features in other tissues during further evolution (Bekpen et al. 2018). In *Drosophila*, highly conserved HOX genes that set up the anterior–posterior body axis have a different function in subsequent development, for example, regulating motor neuron differentiation that determines the behavior of the animals (Joshi et al. 2022). Similarly, dorsal–ventral polarity is established by a genetic pathway starting with the ligand receptor *Toll* that also regulates *Drosophila* immunity, and *Toll*-like receptors are well-known as critical players in human inflammatory and immune disease (Lemaitre et al. 2012).

The multiple functions of genes are well-known under the term "pleiotropy," defined as one polymorphism in a given gene affecting multiple traits (Mackay and Anholt 2024). Human traits are almost invariably driven by pleiotropic gene functions (Solovieff et al. 2013). Genome-wide analyses of polymorphic variants on molecular and organismal phenotypes have revealed the prevalence of pleiotropy among quantitative traits, including diseases (Mackay and Anholt 2024). Human genetic studies now commonly report a web of genetic correlations across say immune or psychiatric diseases, often with other traits (Chesmore et al. 2018). What does this say about our foundation of genetics? Should the overwhelming evidence for pleiotropy of most genes not have long been considered as a severe problem in the idealistic "one gene —one trait" assumption?

## The genetic background issue

It is well established that the effect of genes is modulated by the epistatic interaction with other genes in the genome, and therefore, the genomic background in which a particular mutation is induced ultimately determines which function is ascribed to the gene. The implications of such background effects are regularly not addressed in classical genetics approaches.

Johannsen´s concept of the gene implied that hereditary functions of a gene could be revealed when one manages to experimentally control for other factors influencing the variation, most notably variable genetic backgrounds. This is why he emphasized that isogenic strains are required for doing "exact genetics," a dogma that still holds. But if one tests the same mutation in different isogenic backgrounds, one will very often encounter different phenotypic consequences. In a systematic study with testing the phenotypic effect of a given mutant gene in 30 laboratory strains of mice, it was found that the genetic background could support diametrically opposing conclusions regarding the function of the gene (Sittig et al. 2016). In human genetics, it is well established that the same mutation in a given gene can lead to very different disease phenotype penetrance and expressivity (Kammenga 2017; Kingdom and Wright 2022), apparently depending on the genetic background of the respective individual. It is even possible to find perfectly healthy individuals to carry homozygous null alleles of genes that cause a severe disease in most other individuals (Chen et al. 2016).

Ironically, this is the very problem that experimentalists try to control. By using well-defined genetic model systems, one strives for reproducibility, but the seeming exactness that is achieved with inbred strains thus becomes an illusion contingent on the exact experimental conditions. In many cases, "exact" gene functions exist only for the artificial genetic experiment, but they can look different in other experiments.

## The missing heritability issue

The expectation of the classical genetic approach that gene mutations will reveal the genetic regulation of phenotypes has certainly proved to be true in the context of "making or breaking" a trait. However, when it comes to explaining phenotypic variation across individuals, mapping approaches that do not rely on induced mutations have shown that this assumption is greatly oversimplified.

The genomic revolution made it possible to approach the function of genes in a fundamentally different way. Rather than looking at the effects of knockouts, one can now trace the effects of natural variation represented by different alleles, or polymorphisms, of all genes in the genome at once. This is what is done in GWAS. Using dense SNP information across the whole genome in many individuals for which a quantitative phenotype (e.g. human height) has been recorded, one can identify loci that have an effect on the phenotype, being limited only by the robustness of the phenotype to the environment and the number of individuals investigated. Potentially causal loci show up as significant

association peaks in genomic locations defined by the associated SNPs. In most cases, there are many peaks for a given trait, supporting the general idea of a set of regulatory or coding variants jointly influencing the phenotype. However, even the first systematic studies based on this scheme revealed a sobering problem: the cumulative effect of the variants significantly associated with the trait explained only a fraction of the heritable portion of the phenotype (Manolio et al. 2009).

The now generally accepted explanation for this observation is that this is the result of limited statistical power to identify all relevant associations. In studies that do not include a sufficient number of individuals, many variants contributing to the trait remain under the statistical radar. But some studies now reach the threshold where one can say with reasonable confidence that all common variants that contribute to the phenotype were detected. The most notable one is a meta-analysis of human height GWAS, including data from 5.4 million individuals (Yengo et al. 2022). The authors claim that this study has likely uncovered all common variants that contribute to genetically determined size differences in European-ancestry people. They found over 12,000 independent SNPs in over 7,000 genomic segments representing 21% of the genome. Each single SNP has only a very small positive or negative effect on the height of an individual. This is a typical result for most GWA studies on quantitative traits. In fact, one can even propose that practically every gene that has a segregating variant and is expressed in the relevant tissue or developmental stage contributes to a given quantitative phenotype—the so-called omnigenic model (Boyle et al. 2017). Hence, the discreteness of phenotypic and genetic states that was so fundamental for developing today´s concepts of heredity is dissolving into an almost complete continuity of "infinitesimally" small effects, generated by very many segregating natural genomic variants of the contributing genes.

## What are the challenges in embracing the complexity of genotype–phenotype relationships?

The discovery of the highly polygenic nature of organismic phenotypes has come as a surprise to many biologists who had embraced the classical genetics view and who had hoped that the generation of the phenotype would be driven by manageable numbers of major effect genes only. While human GWAS almost always recover—besides the many minor-effect loci—a few major effect loci, the causative variants are usually rare in the population and probably under negative selection (Wainschtein et al. 2022; Yengo et al. 2022). Although the evidence for high polygenicity does not contradict the notion that discrete Mendelian variants are at the basis of heredity and can in some cases trigger clear phenotypic changes (Bannasch et al. 2021; Enbody et al. 2023; Pallares et al. 2017; Tian et al. 2024), the limitations of the reductionist approach of studying single genes to understand the generation of the phenotype are increasingly apparent.

But what are the challenges? Consider, for example, from a polygenic perspective, the immensely intricate problem to understanding how the shape and function of the vertebrate skull is generated. A functional skull is the end result of remarkably precise coordination of sensory organ development in which the eyes, ears, and nose are seamlessly incorporated into co-functioning units, and where the feeding apparatus is fine-tuned for optimal processing of the available food, all the while also supporting vocalization. Simultaneously, genetic variants exist within any natural population that subtly alter skull shape, yet most allelic

combinations still result in a functional structure, even in hybrid zones between taxa where very different genetic architectures blend into each other (Pallares et al. 2016). Additionally, the skull serves as a key target for adaptive evolution, easily evolving to accommodate new dietary needs, sensory demands, or sexually selected traits. The polygenic model now posits that these diverse phenotypic traits rely on a broad and overlapping network of genes and alleles. How do they maintain this delicate balance between precise integration and evolutionary flexibility?

The discovery of the importance of polygenic genetics makes the mechanisms for establishing the genotype–phenotype map even more mysterious than hitherto appreciated. What are the consequences of understanding how genetic variation is maintained in populations, and how does the pleiotropy exhibited by thousands of variants influencing every trait impact our understanding of the roles of selection and drift (Simons et al. 2018)? The development of a new (old) genetics (Tautz et al. 2023) constitutes therefore one of the largest challenges in current biology. This requires the development of new theory that goes beyond the treatment of genetic loci as abstract entities and incorporates the knowledge about e.g. the molecular properties of genes, their epistatic interactions, and their position in gene regulatory networks. It is equally a challenge for geneticists, as well as for developmental and evolutionary biologists.

## Example studies

To illustrate the types of insight that can be obtained with a molecular quantitative genetics approach, we highlight three studies published just in the past few months. The first has finally molecularly characterized all 7 of the loci used by Mendel in his studies of peas and has identified specific loss-of-function variants for them (Feng et al. 2025). In parallel, however, the same study analyzed 72 additional traits which revealed an oligogenic to polygenic architecture. This contrast shows most directly that the single-gene genetics that is based on Mendelian approaches reveals only a partial insight into the generation of the phenotype.

A study on human bipedalism used a combination of histology, comparative and functional genomics, and bioinformatics to show that highly polygenic effects underlie heterotopic and heterochronic shifts in ossification of the pelvic ileum (Senevirathne et al. 2025). These changes were critical for the evolution of bipedalism and facilitation of a highly modified birth canal. Integration of signals across hundreds of loci provides insight into the phases of directional polygenic selection followed by stabilizing selection on the pelvic structure (Young et al. 2022).

Genome-wide approaches can also be used to illuminate the mechanism of activity of core genes, as shown in Zebell et al. (2025) in their demonstration of how cryptic variation and hierarchical epistasis contribute to the evolution of inflorescence in tomatoes. These examples clearly illustrate the roles of theoretical and evolutionary approaches in the elucidation of novel functional insights, even though there is still a long way toward the mechanistic understanding of these processes.

## Experiments vs description

The development of GWAS over the last two decades has led to the identification of genes and loci underlying polygenic architectures that otherwise had been treated for decades in an abstract form (Abdellaoui et al. 2023; Visscher et al. 2012, 2017). Although the field of complex trait genetics reaches far beyond humans, it is research in this species that currently leads the way. In humans, GWAS are increasingly important in predictive health and personalized genomic medicine, holding in principle the promise of

eventually linking fine mapped causal variants to mechanistic processes (Lappalainen and MacArthur 2021). The motivation is not only medical, namely, identifying the causes of human genetic diseases and/or optimizing therapeutic intervention, but also explicitly about defining the architecture of complex traits. However, studies in humans are naturally only descriptive and provide only very limited options for experimental control of the associated genetic and environmental conditions. As a result, inferences made from human GWAS are incomplete in that regard, despite the fact that it is well acknowledged in the human complex trait community that GxE effects play a critical role in the understanding of genotype–phenotype mapping (Boye et al. 2024; Motsinger-Reif et al. 2024). It is also becoming clear that disease risk evolves within and among populations and is strongly shaped by lifestyle and socioeconomic conditions with sex differences that also highlight context specificity (Zhu et al. 2023).

Human genetics is struggling with the implications of polygenic inheritance for investigating the joint contributions of rare and common variants to complex disease phenotypes (Gibson 2012). Clinical geneticists continue to concentrate on the occurrence of rare single-gene effects as these are seen to be more medically actionable, often with better insights into biological functions. Consequently, even though more people are at risk for many complex common diseases due to their overall genetic profile rather than rare variants (Khera et al. 2018), a consequence is that the classical Mendelian paradigm is again being harnessed, albeit utilizing new genomic analyses (Ropers and van Karnebeek 2022). Others accept that in a polygenic situation, one cannot aim to understand the functional contribution of each allele (Crouch and Bodmer 2020; Gibson 2019; Lambert et al. 2021; Wray et al. 2013). Recent developments like polygenic risk scores tackle the question by modeling many variants of small effect together and can be integrated with metadata to reveal the importance of environmental interactions (Herrera-Luis et al. 2024; Nagpal et al. 2022).

Recently, the IGVF Consortium proposed an ambitious research program for studying the impact of genomic variation on function (IGVF-Consortium 2024). The initiative builds on a growing desire to move from variant to function in human disease research (Lappalainen and MacArthur 2021). IGVF envisages combining single-cell profiling, genome editing, and predictive modeling to catalog the contributions of coding and regulatory variants to cellular and organismal phenotypes. The paradigm represents a contemporary version of molecular genetics and is reductionist in the assumption that variant function can be understood at the individual variant level. Hence, while this will yield a huge amount of very valuable correlative and descriptive data, its main focus will be on understanding disease phenotypes caused by pathogenic alleles, but we also need to develop a fundamental experimental paradigm for understanding how genotypes generate normal phenotypes through naturally segregating alleles, in particular for adaptive and life-history traits.

## Questions that cannot be addressed studying one gene at a time

With these considerations in mind, we conclude this section by outlining five types of question that are representative of the mechanistic gap left by approaches that interrogate biological systems one gene at a time. (i) Are there phenotypes that only arise due to mutations in particular environments, genetic backgrounds, or other exposure contexts? And if so, what are the molecular mechanisms that mediate the uncovering of such cryptic genetic variation? (ii) Are there pathways and networks whose contribution to phenotypic variation can only be revealed by aggregation of hundreds or thousands of small effects on chromatin accessibility or transcript or protein abundance and activity, which cannot be revealed by classical mutagenesis that focuses on single or few genes? (iii) How can we reconcile the observation that GWAS associations that map genotype to disease risk or quantitative traits are very often different from the eQTL signals that explain a large fraction of the expression of the presumed causal gene, residing at different locations in the genome and experiencing different selection (Mostafavi et al. 2023)? (iv) How do gene regulatory networks absorb mutational and environmental perturbations, contributing to phenotypic stability, but genetic variation in the same networks is the raw material for phenotypic evolution? (v) When considering genotype–environment interactions (GxE), what fraction of these effects have a direct mechanistic explanation such as engagement of a glucocorticoid or pattern recognition receptor, what fraction is due to the indirect systemic influence of stressors and behavioral change, what fraction represents modified allelic effect sizes and frequencies among study populations in different environments, and what fraction is simply due to statistical artifacts of processes like scaling and normalization (Westerman and Sofer 2024)? Addressing these and more questions requires experimental systems that engage natural variation, diverse environments, high-throughput phenotypic and multiomic measurements, and polygenic modeling.

## What experimental approaches explicitly address the complexity of genotype–phenotype mapping?

Here we outline approaches that can be used to eventually unravel the mechanistic principles behind polygenic genetics. This is not an exhaustive list, but these approaches are starting points offering a wider perspective on the problem. They capitalize on the power of using natural variation, and they are based in an evolutionary perspective. Each of them can be developed into many new directions, when the accumulating insights generate new ideas for further experimental designs.

### Parallel selection experiments

Aside from GWAS, artificial selection experiments are a tractable option to assist in unraveling the genetic basis of complex traits (Barghi and Schlötterer 2020; Chan et al. 2012; Long et al. 2015). This type of experiment has been performed for decades, but for the most part they have either tracked only the phenotypic changes or only the genomic changes. To link genotype to phenotype, it is required that both, genome and phenotype, are quantified at the beginning and end of the experiments and ideally at several time points in between and in multiple parallels. For example, by comparing the genomic changes at the endpoints of parallel selection lines for larger body size in mice, it was possible to disentangle the genetic pathways underlying the selected trait by genotyping just a handful of individuals (Chan et al. 2012). Of course, all the experimental effort in this case was in the decades-long selection regime, but it should be possible to work with much fewer generations of selection by utilizing more parallel lines and careful choice of the genetic composition of the starting population to reduce linkage concerns. Explicit simulation of such breeding programs can aid the design of these experiments (Gaynor et al. 2021; Kofler and Schlötterer 2014). Furthermore, by tracking evolution in real time, it is possible to uncover which molecular pathways are responsible for trait variation. The parallel-lines setup allows for testing whether the genetic trajectories are repeatable or not (Schlötterer 2023), thereby providing also insight into

mechanisms of redundancy and compensation. If in addition to morphological traits, gene expression or chromatin organization data are also collected, it will be possible to quantify whether and how gene regulatory networks are rearranged to generate novel phenotypes (Halfon 2017). Parallel selection experiments therefore offer a window into how the genome changes in a concerted way to generate fitter phenotypes that are better adapted to new environments.

A special advantage of the parallel selection approach in model systems is that the starting population can be modified or chosen in a way that specific questions can be addressed explicitly (Barghi et al. 2020; Schlötterer 2023). For example, what is the effect of having more or less linkage-disequilibrium in the starting population? What are the effects of the population size or different selection regimes? What are the effects of different environments? Once specific pathways have been identified experimentally, targeted mutagenesis on whole pathways can be used to gain a more mechanistic understanding of the impact of genetic variation in such pathways, e.g. (Li et al. 2024). Alternatively, experimental designs that compare the result of selection on randomly mutagenized lines versus naturally segregating variation can be used to better understand the role of new mutations versus standing genetic variation in shaping adaptive phenotypic variation.

To conduct these types of experiments in a powerful way, i.e. with many parallel lines and with different starting conditions, will require experimental dimensions orders of magnitudes larger than currently performed. Many replicates subjected to selection under controlled conditions are necessary to distinguish random from true biological signal in these experiments. For example, even a single parallel selection experiment may involve more than 10,000 individuals that need to be phenotyped and genotyped. Rapid progress in genomics has made the sequencing of large numbers of genomes already more accessible. However, developing high-throughput phenotyping tools will require in addition also new sophisticated engineering, combined with advanced image analysis and computational techniques.

### GWAS in environmental perturbation experiments

The effect of the environment on polygenetic inheritance, including the genetic control of variance, can be studied by controlled environmental perturbation experiments. Such experiments can incorporate physical, chemical, and genetic perturbations. Large-scale GWAS under multiple controlled and replicated conditions is becoming feasible with classical experimental model systems (e.g. *Drosophila*, *Arabidopsis*, zebrafish, *C. elegans*, yeast), as well as emerging ones. This approach will further our understanding not only of the genetic basis of complex traits, but will also shed light on the molecular pathways that are more or less robust to genetic and/or environmental perturbation and how regulatory networks adjust to cope with new environmental circumstances. The answers will greatly impact the way we understand how functional traits respond to the dynamic environment in which organisms live, with implications for conservation genetics and modeling response to climate change.

### Parallel evolution in natural populations

Parallel selection has occurred in a variety of natural populations. It has been observed especially in the wake of the recolonization of empty habitat patches, for example, after the last glaciation. If the colonizers were derived from known, more-or-less well-defined source populations, one can treat them as natural experiments that can be harvested to understand the underlying genetics.

A text book example are the three-spined sticklebacks, which colonized freshwater from the ocean in many localities in the Holarctic (Jones et al. 2012). Other examples include the marine snail *Littorina* along shore gradients (Morales et al. 2019), water-fleas adapting to brackish water (Santos et al. 2024), *Arabidopsis arenosa* adapting to elevational clines on different mountain ranges (Knotek et al. 2020), or weed populations adapting to the intensification of agriculture (Kreiner et al. 2022).

Phenotypic response to selection in the new environment is often fast, with selection acting on standing genetic variation from the source population (Fuhrmann et al. 2023). The genomic architecture associated with parallel phenotypic evolution is often characterized by genetic trajectories, impacting multiple genes. Often these trajectories are observed to evolve in parallel, while others are specific to individual populations (Bradic et al. 2013; Holliday et al. 2016; Jones et al. 2012; Manceau et al. 2010; Soria-Carrasco et al. 2014). The more similar the source populations are, the stronger the case for parallelism. Likewise, the more similar the ecological conditions, the stronger the case for parallelism.

Harvesting examples of parallel evolution in natural populations has only recently become more widely addressed and used as a tool for evolutionary genetics, but certainly many more natural systems can be used for this approach and high numbers of replication may be possible. In a similar vein, we can exploit human-made changes to the environment as natural experiments, e.g. moth adaptation to light pollution, weeds adapting to agricultural practices, fish adapting to fishery practices, birds and mammals adapting to urban environments, industrial melanism in insects, and aquatic organisms adapting to stressors in lakes and rivers.

## Conclusion

Despite the fact that the apparent conflict between heredity being based on particulate elements or continuous variation was resolved at the beginning of the last century, genetic analysis continues to be dominated by approaches centered on the function of single genes. By contrast, quantitative genetic approaches increasingly emphasize the central role of phenomena such as polygenicity, context-dependency, epistasis, pleiotropy, and redundancy. Here we call once again for the community of researchers that engages in more classical genetics to also embrace the complexities of genotype–phenotype mapping and ask them to appreciate the need for the development of experimental genetic approaches that assess mechanism through the lens of polygenicity.

To enable this paradigm change, we call for new initiatives, including new institutes and research centers deliberately charged with concentrating efforts on large-scale experimental approaches such as those delineated here. A unique aspect of the infrastructure for such institutes, not typical for current centers for genomics, would be engineering units that develop advanced automated phenotyping procedures. These would have to be specifically designed for every major experiment, and they would therefore in themselves constitute a scientific and engineering challenge. We are now called not only to move from studying one gene at a time but also to move from observation to experimentation while embracing biological complexity.

## Acknowledgments

We thank a variety of further colleagues for the many inspiring discussions on the nature of heredity, especially the workshops in Berlin. Special thanks also to the Stellenbosch Institute for Advanced Studies (STIAS) to provide DT the leisure and freedom

to write up the first version of this perspective. Thanks also to three reviewers who have helped to improve the manuscript.

## Funding

Two dedicated symposia on the topic were funded by the Max-Planck Society.

## Conflicts of interest

None declared.

## Author contributions

This perspective is based on discussions, published articles and two symposia ("Complex Trait Genetics" and "Rapid Polygenic Adaptation" at the Harnack House in Berlin). The core text was written by Diethard Tautz with major additions from Greg Gibson, Luisa Pallares, and Dieter Ebert; extensive text edits were provided by Detlef Weigel and Didier Stainier. All further authors were involved in at least one of the above activities, and several have added comments and corrections that have shaped the final text.

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

*Editor: A. Long*