## [Peer Review File · Genetics]

Beyond Mendel: a call to revisit the genotype-phenotype map through new experimental paradigms

Diethard Tautz, Luisa Pallares, Andersson Leif, Neda Barghi, Nicholas Barton, Bay Rachel, Chan Frank, Hancock Angela, Kaiser Tobias, Daniel Koenig, Zacharias Kontarakis, Liedvogel Miriam, DeMeaux Juliette, Magnus Nordborg, Palmer Abraham, Michael Purugganan, Christian Schlötterer, Karl Schmid, Didier Stainier, Weigel Detlef, Jochen Wolf, Dieter Ebert, and Greg Gibson

NOTE: The reviews and decision letters are unedited and appear as submitted by the reviewers.

In extremely rare instances and as determined by a Senior Editor or the EIC, portions of a review may be redacted. If a review is signed, the reviewer has agreed to no longer remain anonymous.

The review history appears in chronological order.

Review Timeline:

Submission Date:	2025-07-09
Editorial Decision:	2025-08-27
Resubmission Received:	2025-11-18
Editorial Decision:	2025-12-22
Revision Received:	2026-01-13
Accepted:	2026-01-17

August 26, 2025

GENETICS-2025-308353

The need to revisit the mechanistic basis of phenotypic inheritance through new experimental paradigms

Dear Dr. Tautz:

Two experts in the field and a more junior reviewer (#1) have reviewed your manuscript, and I have read it as well. The review is very timely, and there was considerable enthusiasm for the idea of revisiting the genetics of complex traits. The reviewers thought there were positives but also had major problems with the manuscript, as written. I believe considerable work is required to make it acceptable for publication. While your manuscript is not currently acceptable for publication in GENETICS, I would consider a substantially revised manuscript. Both reviewers have comments and concerns to be addressed in a revised manuscript. You can read their reviews at the end of this email.

I feel there were three big issues with the paper that I would need to see much more fully addressed.

1a. There is a rich, long history of work that has embraced polygenicity & complexity from the very beginning that has produced a lot of valuable knowledge about the emergent properties of genetics (if not about specific gene mechanisms). The coverage of this work was in many places superficial or lacking. It is odd as at least one of the authors (Barton) has made seminal contributions to this literature. More work is required in terms of what classical theory has already told us.

1b. Perhaps part of the problem is that the authors are setting up a "straw man" dominant view as that of molecular biologists or developmental geneticists ("Mendelins"). But many of us have taught courses for years using Falconer and Mackay as the textbook (certainly anyone at an ag school). And some of the surprises of this review are textbook stuff. The idea that selection can move phenotypes well outside the range of normal variation is obvious to a chicken or pig breeder, or anyone who has a casual understanding of dog breeds. There is an entire section of Genetics devoted to "complex traits". A perspective in Genetics needs to address practitioners of the field of The Genetics of Complex Traits.

2. For a review of where the field is and where it has to go published one quarter of the way through this century, it is oddly silent on human GWAS. Clearly this work has had an outsized impact on where we are and what we are thinking (even if I agree there is an over-emphasis on the idea that this is the entire picture). But more engagement is needed here. What is it that human GWAS cannot tell us and why there remains a need for model system work.

3. Both reviewers and I felt the review ended rather abruptly and specifically saying the answer to the questions in the field are evolve and resequence experiments. Clearly I support this work, and have even participated to some degree. But, this was strange as the review doesn't really show that this field has made seminal contributions to current thinking, and even some of the authors on the paper do not really believe this is the only or even main approach we should be taking moving I suspect (If I am wrong I want signed receipts from Hancock, Nordborg, Palmer & Purugganan). I am excited about E&R, but the review may be better served by suggesting a few ways forward. Or if E&R really is the only way forward and this is what the authors think, be more specific about why it is the only way forward, and why other approaches are so limited and not going to tell us. Why is this the way forward and what are the deliverables?

We look forward to receiving your revised manuscript. Please let the editorial office know approximately how long you expect to need for revisions.

Upon resubmission, please include:

1. A clean version of your manuscript;
2. A detailed response to the editor's/reviewers' feedback and to the concerns listed above.
3. I feel the revisions are significant enough that some sort of track changes document would reflect not taking the reviewer's concerns to heart.

Your paper will likely be sent back out for review.

Additionally, please ensure that your resubmission is formatted for GENETICS
<https://academic.oup.com/genetics/pages/general-instructions>

Follow this link to submit the revised manuscript: Link Not Available

Sincerely,
Anthony Long
Senior Editor
GENETICS

Approved by:
Howard Lipshitz
Editor-in-Chief
GENETICS

Reviewer #1 Comments:

The paper outlines a well-written and much needed perspective in the study of complex trait genetics. This is an accessible yet substantive read for early career scientists, researchers in sister disciplines, grant agencies, and the general public. Through a historical lens, It evaluates and summarizes why certain approaches have become more preferred over others- for e.g. the focus on single gene despite Fisher's integration of two schools of thoughts in 1918, and its implications for the field so far and what the next steps should be. Further, the perspective offers approaches that are needed for the field to move forward in understanding mechanisms beyond a narrow context in model organisms. This is particularly valuable for not only understanding complex biology but also to enhance likelihood of biological discovery being immediately applicable, which likely is in line with many funding agencies.

There are no major issues, however the new experimental paradigms aren't sufficiently fleshed out. Few suggestions are listed:

Line 425, In "...here one starts with outbred populations..." This makes sense but expanding the consequences of starting population composition could be valuable here. For e.g. what happens if one uses a diversity panel in a crop like maize or a randomly mutagenized population of an elite variety, etc. in the same vein, how can existing knowledge about these populations be used to assess if the question being asked can be answered with that population

Line 445-450, "...it should be possible to work with fewer generations of selection... "bread and butter". This is a great point and to make sure this message is communicated to the researchers and grant reviewers, being able to simulate what one would do experimentally insilico would be of great value. There are already tools in Population genomics or in breeding like AlphaSimR that likely do some or all of it. Encompassing those discussions in this context could facilitate the much needed discussion in the field and help people see the predictions before the experiments. There is a risk of people classifying many of these experiments as intractable. The authors briefly mention these and other ideas in the Box1 but the extent of discussion may not provide enough action items or resources to actually get a picture of what the experiment might look like.

One issue to complex traits is the effect size and the population needed to detect such effect sizes tend to be large. Further discussion on how such challenges could be addressed in the new paradigm is worth more discussion.

Reviewer #2 Comments:

The Perspective article by Andersson et al., "The need to revisit the mechanistic basis of phenotypic inheritance through new experimental paradigms", argues that the Mendelian, one-gene-at-a-time approach to understanding the genotype-phenotype relationship is insufficient for understanding complex phenotypes and that new experimental approaches (parallel selection experiments) are needed to "understand the mechanisms underlying phenotypic heredity".

I do not feel that this article presents a scholarly viewpoint about the current challenges and prospects for understanding the genetics of quantitative traits, nor an adequate defense of their proposed new experimental paradigm. In fact, the authors barely mention that the 'phenotypes' with which they are concerned are quantitative traits. They begin by arguing that all of genetics until now has been based on single gene mutagenesis and transgenesis studies in inbred genetic backgrounds. It is certainly true that such studies have been more common than the study of quantitative phenotypes, but the authors ignore (or are not aware of) pioneering work on theoretical and experimental work on the genetics of quantitative traits. Quantitative genetics theory begins with Sir Ronald Fisher's famous 1918 treatise, "The correlation among relatives on the supposition of Mendelian inheritance". This paper reconciled differences between the biometricians (Galton, Pearson) and the Mendelists (e.g., Morgan). It is the foundation of quantitative genetics, explaining that continuous variation in quantitative traits (the 'phenotypes' relevant to this Perspective) arise from the joint segregation of variants at many individual genes, each with a small effect on the trait, together with effects of the environment (all non-genetic causes of variation). Fisher generalized the concepts of Mendelian inheritance including co-dominance (single locus additive effects); dominant/recessive gene action, pleiotropy (via genetic correlation) and epistasis (which he called epistacy). He expressed these concepts in terms of variation in populations, confounding effects with allele frequency. However, the basic equations he derived for additive, dominance and epistatic effects and variances are based on single (or two, for the case of epistasis) loci. The conclusion of this Perspective states: "After more than hundred years of genetic research, the rather disruptive discussion of whether heredity is based on particulate elements or continuous variation has been resolved: it is both!" Fisher's paper reached the same conclusion in 1918.

Although researchers in the middle of the 20th century did not have the tools available today to identify the variants and genes affecting quantitative traits, they did utilize selection experiments in model organisms to test quantitative genetic theory and begin to map the first 'polygenes' by linkage to visible polymorphisms. It was quickly realized that model quantitative traits such

as mouse body weight and abdominal and sternopleural bristle numbers in *Drosophila* were highly polygenic since selection moved the population means well outside the range of variation in the original populations. Theory developed on selection limits and the interplay of selection, drift and linkage during artificial selection remain relevant today, as does theory on the role of new mutations in selection response. The authors claim that one of the outcomes of their proposed research agenda will be to develop the theory of polygenic selection. What does this theory need to do beyond what has already been developed? The authors appear to think that polygenic variation is something new: "Polygenic genetic variation has turned out to be a sufficient basis to shape phenotypes in almost any direction and background variability is essential to overcome deleterious effects that accompany genetic engineering. It usually does not require new mutations or the introduction of artificially manipulated genes to achieve the goals." "The discovery of the highly polygenic nature of the generation of organismic phenotypes has come as a shock even to many specialists." Perhaps the magnitude of polygenicity is surprising (i.e., thousands rather than perhaps hundreds of contributing loci), but hardly the concept itself.

What is missing from this article is a clear articulation of what problems remain to be solved to "understand the mechanisms underlying phenotypic heredity". The authors do give some arguments, but they need to be fleshed out. What about environmental effects and genotype by environment interactions need to be addressed? Given that environmental effects are, by definition, non-genetic; how are they relevant in this context? Other issues raised include various context-specific effects that are the result of epistatic interactions (e.g., naturally occurring modifiers that enhance or suppress the effects of mutations revealed by studying the same variants/mutations in different genetic backgrounds) and pleiotropy as well as the need to understand how variants/genes associated with quantitative traits work in networks. However, details are vague. For example, what is the nature of these networks? Transcriptional? Protein-protein interactions? Gene-gene interactions? All of the above?

In the absence of the articulation of explicit issues to be solved, it is difficult to understand how the proposed 'novel' experimental paradigm - replicated selection experiments in model organisms. As noted above, artificial selection for quantitative traits is not new. What is new is the realization of the extent of polygenicity, the need for huge sample sizes per replicate and high replication to attempt to tease apart the effects of drift and selection. Whole genome DNA sequencing of pools of individuals from these populations over time enables tracking allele frequency changes accurately in a controlled laboratory environment, and phenotyping large numbers of individuals enables assessing which alleles are associated with the phenotype under selection. Such experiments will yield a list of variants/associated genes with consistent changes in allele frequency over time. How does such a list differ from the list that would be obtained by genotyping and broadly phenotyping thousands of individuals in the starting population - i.e., a GWAS? How exactly does this method address the issues of pleiotropy, genotype-environment interaction, epistasis and network biology that need to be solved?

I agree that experiments using populations of genetically polymorphic model organisms in different controlled environments have a lot to add to our knowledge of the genetic architecture of quantitative traits. Larger samples always give more power, whole genome DNA sequencing gives a complete picture of genetic variation, especially if long reads are employed, and automated phenotyping of multiple traits on the same individuals will resolve pleiotropy. The addition of RNA sequencing and other 'omics analyses of dissected tissues and at different times of development would also provide the opportunity to understand how variants are associated with the phenotypes via various network levels (i.e., systems genetics). However, there are experimental designs other than parallel selection experiments that can achieve the same goal, and the disadvantages of the parallel selection paradigm are not addressed.

In conclusion, the authors fail to outline what "mechanisms underlying phenotypic heredity" need to be discovered by the new field of "experimental polygenic genetics". This is a Perspectives article, so the authors have leeway in expressing their opinions. However, rather than clearly outlining challenges and potential solutions, the paper becomes a plea for funding their own research programs.

Reviewer #3 Comments:

Andersson et al. make an argument that the field of genetics needs to more fully embrace the complexity of most phenotypes and that reaching this goal will also require a shift to new experiments. They provide the historical context for this argument and highlight several areas of genetics where major gaps in knowledge remain. I wholeheartedly agree that an overly simplistic view of the genetic basis of phenotypes has been too prevalent in the field of genetics and that this perspective being entrenched in the field has limited the advancement of knowledge. Thus, I do find this perspective to be valuable and needed in the field. However, I also have some suggestions for the framing of these issues and some concerns in the current draft that I detail below.

1) Increased acknowledgement and discussion of those in the field who have embraced complexity in genetics

On line 356, the authors write, "The discovery of the highly polygenic nature of the generation of organismic phenotypes has come as a shock even to many specialists." Not just in this one section, but throughout the manuscript, I felt the tone was that pretty much all geneticists shared the simplistic view of genetics they describe. I think it is important to point out that 1) in fact many people (probably nearly all scientists who describe themselves as quantitative geneticists) were not at all surprised, 2) that arguments emphasizing the importance of recognizing the likely highly complex nature of the genetic basis of most phenotypes have been made before several times (a couple examples include: Rockman 2012 *Evolution*; Travisano & Shaw 2013

Evolution), and 3) that there is a rich, long history of work that has embraced polygenicity & complexity from the very beginning that has produced a lot of valuable knowledge about the emergent properties of genetics (if not about specific gene mechanisms).

I understand the decision to emphasize and spend the most time detailing the viewpoint that is being argued against here. However, I think it is equally important to point out that there has been an alternative viewpoint and well-developed theory that predicted high complexity. While the authors do acknowledge Fisher's infinitesimal model and the later omnigenic model, I did find their coverage of the history of the field to be uneven with a strong emphasis on the simplistic view. I was also surprised that the modern synthesis was not discussed more. The authors describe the divide that existed before the modern synthesis but don't fully explain how the first time the field realized: "whether heredity is based on particulate elements or continuous variation has been resolved: it is both!", as they state in the conclusion, was during the modern synthesis in the 1930's and 40's (Provine's book, *The Origin of Theoretical Population Genetics* is a nice discussion of this history). It may be useful to point out more explicitly that attempts to bring these perspectives together have been made repeatedly throughout this history but there has been continual resistance to a full acknowledgement and embrace of complexity.

2) Clarification of what exactly the field needs to "revisit."

The authors use a couple different phrases that they argue the field needs to revisit, including "the mechanistic basis of phenotypic inheritance" and "the mechanisms of heredity". I think if you say those phrases to different people, you will get different answers about what they think they mean. I also think that for many, the phrase will imply that the argument is suggesting that the fundamental misunderstanding that is being discussed has to do with how individual alleles are inherited and/or some other aspect of the mechanisms of gene action. However, I don't think this is actually what the authors think needs changing. I would argue that there is not anything about mechanism that is not completely compatible with a model of most traits being highly polygenic and context dependent. Isn't the argument more about an increased openness to ideas like: traits being determined by many, many variants and the effects of a given variant typically being context dependent? From my perspective, these ideas don't challenge the fundamental mechanism of how variants are inherited but are instead different hypotheses about the characteristics of the variants underlying a given phenotype. I think the perspective would be more effective if it stated very clearly what it means (and does not mean) by these phrases. Similarly, the authors use "Mendelian" as a short-hand for the simplistic perspective they are arguing against but elsewhere note that they are not arguing against the idea that "discrete Mendelian variants are at the basis of heredity." So, while I think I understand the perspective they mean when they say, "Mendelian", I don't think it is the best term to encompass the viewpoint given that we use this as a way to accurately describe how individual variants are inherited. I would encourage the authors to spend some time fully defining the perspective they are arguing against (e.g. "The dominant perspective in the field of molecular genetics has implicitly assumed that 1) 2).... 3)....") and give this perspective a name that is not also the name of a legitimate process not in question so that you can refer to it easily throughout but also be sure that the reader has the same understanding of what is meant.

3) More careful consideration of what needs citation and which papers to cite.

I am sympathetic to the scope of the ideas being presented here and the sheer number of references that could be cited for a given idea. However, I do think there are fewer citations than needed and that in some cases fundamental references are missing. I'd encourage the authors to go back through and assess where they have statements that need a citation but have none. This is a general comment but a couple of examples of what I mean are below.

On line 155, they introduce the idea of a GxE, a concept with a very long history with many papers and books discussing its definition, potential role, and research findings. They choose to cite a single, relatively recent review paper here. I felt this choice left out many important citations and also fails to show the volume of work in this area that already exists.

Both the history section and the section on new paradigms especially seemed to have many statements without a citation that I think need some. For example, "Artificial selection experiments have been performed for decades, hence the general understanding of how they work is well established.", and, "And they need a good number of replicates to allow statistical analysis of allele frequency changes. The latter example in particular should cite the several simulation studies that have demonstrated this fact.

4) A more complete discussion of what the limitations and the potential results are for the new paradigms they suggest.

The authors make an argument for a fairly specific experimental program as a way to move the field forward: experimental evolution studies. However, they are a bit vague on exactly how they imagine the resulting datasets will better inform our understanding of which gene variants contribute to which phenotypes and what the expected effects of these variants are in different contexts. In particular, there is not discussion of potential limitations that will remain or what might be challenging in taking this approach. I am very much in favor of and in support of an experimental evolution approach, but it is also true that there are real challenges to understanding which variants contribute to adaptation using this approach and that many of the realities of polygenicity will also present limitations when using this approach. I think the authors should acknowledge this.

Responses to reviews of the perspective manuscript

Decision text in black, responses in blue

#####

GENETICS-2025-308353

The need to revisit the mechanistic basis of phenotypic inheritance through new experimental paradigms

Two experts in the field and a more junior reviewer (#1) have reviewed your manuscript, and I have read it as well. The review is very timely, and there was considerable enthusiasm for the idea of revisiting the genetics of complex traits. The reviewers thought there were positives but also had major problems with the manuscript, as written. I believe considerable work is required to make it acceptable for publication. While your manuscript is not currently acceptable for publication in GENETICS, I would consider a substantially revised manuscript. Both reviewers have comments and concerns to be addressed in a revised manuscript. You can read their reviews at the end of this email.

>Thank you for getting these reviews which make all very valid points. We have carefully considered them and incorporated them as much as possible, while keeping in mind the readability and the general goals of this perspective.

We respond to each reviewer comment below, but in general we have greatly re-structured the text to make sections more cohesive, and the message of this perspective clearer. We felt that some of the reviewers' comment might have missed the goal of the perspective, and mistook our concise writing for lack of scholarship. We like to emphasize that this manuscript is not a review about a history of genetics, but it is a perspective that aims to highlight the historical contingencies that brought us to the current way of approaching genotype-phenotype relationships and to call the community to embrace the complexities of such G-P map.

We hope that the revised version of this manuscript is able to transmit our message in a clearer way, while paying justice to the fields that have already been engaging with complex traits genetics. To make this clear, we have edited the text to be more explicit about which portion of the genetics community we are trying to address.

I feel there were three big issues with the paper that I would need to see much more fully addressed.

1a. There is a rich, long history of work that has embraced polygenicity & complexity from the very beginning that has produced a lot of valuable knowledge about the emergent properties of genetics (if not about specific gene mechanisms). The coverage of this work was in many places superficial or lacking. It is odd as at least one of the authors (Barton) has made seminal contributions to this literature. More work is required in terms of what classical theory has already told us.

1b. Perhaps part of the problem is that the authors are setting up a "straw man" dominant view as that of molecular biologists or developmental geneticists ("Mendelians"). But many of us have taught courses for years using Falconer and Mackay as the textbook (certainly anyone at an ag school). And some of the surprises of this review are textbook stuff. The idea that selection can move phenotypes well outside the range of normal variation is obvious to a chicken or pig breeder, or anyone who has a casual understanding of dog breeds. There is an entire section of Genetics devoted to "complex traits". A perspective in Genetics needs to address practitioners of the field of The Genetics of Complex Traits.

> We agree that our focus on describing how we got to the one-gene-one-trait view could be understood as if nothing else had been done in terms of quantitative genetics. We have now re-

structured the first section of the manuscript that is now called 'How did we get here?' to include a section on the theoretical and experimental developments that provide the data for polygenic architectures. This section is now called 'Genetics without genes' to serve the narrative where we try to explain how we got to the current view.

We have also edited the wording across the whole manuscript to make it clear that we are trying to talk to researchers that still use 'classical genetics' approaches and not to the whole community of geneticists, which as you point out includes many people and fields that have embraced complex trait genetics for a long time.

2. For a review of where the field is and where it has to go published one quarter of the way through this century, it is oddly silent on human GWAS. Clearly this work has had an outsized impact on where we are and what we are thinking (even if I agree there is an over-emphasis on the idea that this is the entire picture). But more engagement is needed here. What is it that human GWAS cannot tell us and why there remains a need for model system work.

> We have now re-organized the bits that dealt with GWAS into two new sub-sections:
- within the section 'What are the challenges in embracing the complexity of genotype-phenotype relationships?'
- in the section 'What experimental approaches explicitly address the complexity of genotype-phenotype mapping?', we again discuss GWAS and its limitations in the subsection 'GWAS in environmental perturbation experiments'.

3. Both reviewers and I felt the review ended rather abruptly and specifically saying the answer to the questions in the field are evolve and resequence experiments. Clearly I support this work, and have even participated to some degree. But, this was strange as the review doesn't really show that this field has made seminal contributions to current thinking, and even some of the authors on the paper do not really believe this is the only or even main approach we should be taking moving I suspect (If I am wrong I want signed receipts from Hancock, Nordborg, Palmer & Purugganan). I am excited about E&R, but the review may be better served by suggesting a few ways forward. Or if E&R really is the only way forward and this is what the authors think, be more specific about why it is the only way forward, and why other approaches are so limited and not going to tell us. Why is this the way forward and what are the deliverables?

> In the previous version we had given more relevance to experimental evolution approaches while alternative ideas were just mentioned in brief in Box 1. To avoid that the ideas in box 1 got lost, we have now removed box 1 and created three subsections that deal with promising experimental approaches. We believe that now we offer a more balanced perspective on potential avenues to follow up on.

Reviewer #1 Comments:

The paper outlines a well-written and much needed perspective in the study of complex trait genetics. This is an accessible yet substantive read for early career scientists, researchers in sister disciplines, grant agencies, and the general public. Through a historical lens, It evaluates and summarizes why certain approaches have become more preferred over others- for e.g. the focus on single gene despite Fisher's integration of two schools of thoughts in 1918, and its implications for the field so far and what the next steps should be. Further, the perspective offers approaches that are needed for the field to move forward in understanding mechanisms beyond a narrow context in model organisms. This is particularly valuable for not only understanding complex biology but also to enhance likelihood of biological discovery being immediately applicable, which likely is in line with many funding agencies.

> Thank you for this general assessment, which reflects indeed how we see the intentions of this manuscript

There are no major issues, however the new experimental paradigms aren't sufficiently fleshed out. Few suggestions are listed:

Line 425, In "...here one starts with outbred populations..." This makes sense but expanding the consequences of starting population composition could be valuable here. For e.g. what happens if one uses a diversity panel in a crop like maize or a randomly mutagenized population of an elite variety, etc. in the same vein, how can existing knowledge about these populations be used to assess if the question being asked can be answered with that population

Line 445-450, "...it should be possible to work with fewer generations of selection... "bread and butter". This is a great point and to make sure this message is communicated to the researchers and grant reviewers, being able to simulate what one would do experimentally *insilico* would be of great value. There are already tools in Population genomics or in breeding like AlphaSimR that likely do some or all of it. Encompassing those discussions in this context could facilitate the much needed discussion in the field and help people see the predictions before the experiments. There is a risk of people classifying many of these experiments as intractable. The authors briefly mention these and other ideas in the Box1 but the extent of discussion may not provide enough action items or resources to actually get a picture of what the experiment might look like.

One issue to complex traits is the effect size and the population needed to detect such effect sizes tend to be large. Further discussion on how such challenges could be addressed in the new paradigm is worth more discussion.

> Both of these points are very valid and address the general issue that there are many alternative experimental starting options for such experiments. The goal of the perspective is to get the discussion going, rather than offering already all solutions. We have restructured the section on parallel selection and added also two references to simulating such experiments.

Reviewer #2 Comments:

The Perspective article by Andersson et al., "The need to revisit the mechanistic basis of phenotypic inheritance through new experimental paradigms", argues that the Mendelian, one-gene-at-a-time approach to understanding the genotype-phenotype relationship is insufficient for understanding complex phenotypes and that new experimental approaches (parallel selection experiments) are needed to "understand the mechanisms underlying phenotypic heredity".

> We have now significantly edited the text to incorporate the comments from the reviewer, as well as other reviewers, which makes it hard to highlight in this response all the places where the text was edited or re-organized. But below, we give some examples for each comment.

I do not feel that this article presents a scholarly viewpoint about the current challenges and prospects for understanding the genetics of quantitative traits, nor an adequate defense of their proposed new experimental paradigm. In fact, the authors barely mention that the 'phenotypes' with which they are concerned are quantitative traits. They begin by arguing that all of genetics until now has been based on single gene mutagenesis and transgenesis studies in inbred genetic backgrounds. It is certainly true that such studies have been more common than the study of quantitative phenotypes, but the authors ignore (or are not aware of) pioneering work on theoretical and experimental work on the genetics of quantitative traits. Quantitative genetics theory begins with Sir Ronald Fisher's famous 1918 treatise, "The correlation among relatives on the supposition of Mendelian inheritance". This paper reconciled differences between the biometricians (Galton, Pearson) and the Mendelists (e.g., Morgan). It is the foundation of quantitative genetics, explaining that continuous variation in quantitative traits (the 'phenotypes' relevant to this Perspective) arise from the joint segregation of variants at many individual genes, each with a small effect on the trait, together with effects of the environment (all non-genetic causes of variation). Fisher generalized the concepts of Mendelian inheritance including co-dominance (single locus additive effects); dominant/recessive gene action, pleiotropy (via genetic correlation) and epistasis (which he called epistacy). He expressed these concepts in terms of variation in populations, confounding effects with allele frequency. However, the basic equations he derived for additive, dominance and epistatic effects and variances are based on single (or two, for the case of epistasis) loci. The conclusion of this Perspective states: "After more than hundred years of genetic research, the rather disruptive discussion of whether heredity is based on particulate elements or continuous variation has been resolved: it is both!" Fisher's paper reached the same conclusion in 1918.

> We are copying here the response already given to a previous comment.

"We agree that our focus on describing how we got to the one-gene-one-trait view could be understood as if nothing else had been done in terms of quantitative genetics. We have now re-structure the first section of the manuscript that is now called 'How did we get here? The birth of genetics, and the establishment of the dominant view' to include a section on the theoretical and experimental developments that provide the data for polygenic architectures. This section is now called 'Genetics without genes' to serve the narrative where we try to explain how we got to the current view."

In addition, we have now reworded the sentence mentioned by the reviewer that starts the 'conclusion' section. It now reads "Despite the fact that the apparent conflict between heredity being based on particulate elements or continuous variation was resolved at the beginning of the last century, genetic analysis continues to be dominated by approaches centred on the function of single genes".

Although researchers in the middle of the 20th century did not have the tools available today to identify the variants and genes affecting quantitative traits, they did utilize selection experiments in model organisms to test quantitative genetic theory and begin to map the first 'polygenes' by linkage to visible polymorphisms. It was quickly realized that model quantitative traits such as mouse body weight and abdominal and sternopleural bristle numbers in *Drosophila* were highly

polygenic since selection moved the population means well outside the range of variation in the original populations. Theory developed on selection limits and the interplay of selection, drift and linkage during artificial selection remain relevant today, as does theory on the role of new mutations in selection response. The authors claim that one of the outcomes of their proposed research agenda will be to develop the theory of polygenic selection. What does this theory need to do beyond what has already been developed? The authors appear to think that polygenic variation is something new: "Polygenic genetic variation has turned out to be a sufficient basis to shape phenotypes in almost any direction and background variability is essential to overcome deleterious effects that accompany genetic engineering. It usually does not require new mutations or the introduction of artificially manipulated genes to achieve the goals." "The discovery of the highly polygenic nature of the generation of organismic phenotypes has come as a shock even to many specialists." Perhaps the magnitude of polygenicity is surprising (i.e., thousands rather than perhaps hundreds of contributing loci), but hardly the concept itself.

>In the section 'genetics without genes' we have now added information about the selection experiments on bristle numbers and body size to illustrate the progress made by the quantitative researchers.

We have also edited the text to clarify that it is a specific part of the community that was surprised by the high polygenicity of many traits. This will hopefully make it clear to the reader that we are not referring to quantitative researches, but to people still following a one-gene-one-trait view of genetics.

What is missing from this article is a clear articulation of what problems remain to be solved to "understand the mechanisms underlying phenotypic heredity". The authors do give some arguments, but they need to be fleshed out. What about environmental effects and genotype by environment interactions need to be addressed? Given that environmental effects are, by definition, non-genetic; how are they relevant in this context? Other issues raised include various context-specific effects that are the result of epistatic interactions (e.g., naturally occurring modifiers that enhance or suppress the effects of mutations revealed by studying the same variants/mutations in different genetic backgrounds) and pleiotropy as well as the need to understand how variants/genes associated with quantitative traits work in networks. However, details are vague. For example, what is the nature of these networks? Transcriptional? Protein-protein interactions? Gene-gene interactions? All of the above?

>We have re-structured the text to highlight several of the main challenges in embracing high polygenicity and added the section: 'What are the main challenges in embracing the complexity of genotype-phenotype relationships?'

We also added a short introduction to the 'issues' sections to make it clearer what are the problems that one-gene-one-trait approaches cannot address.

In the absence of the articulation of explicit issues to be solved, it is difficult to understand how the proposed 'novel' experimental paradigm - replicated selection experiments in model organisms. As noted above, artificial selection for quantitative traits is not new. What is new is the realization of the extent of polygenicity, the need for huge sample sizes per replicate and high replication to attempt to tease apart the effects of drift and selection. Whole genome DNA sequencing of pools of individuals from these populations over time enables tracking allele frequency changes accurately in a controlled laboratory environment, and phenotyping large numbers of individuals enables assessing which alleles are associated with the phenotype under selection. Such experiments will yield a list of variants/associated genes with consistent changes in allele frequency over time. How does such a list differ from the list that would be obtained by genotyping and broadly phenotyping thousands of individuals in the starting population - i.e., a GWAS? How exactly does this method address the issues of pleiotropy, genotype-environment interaction, epistasis and network biology that need to be solved?

> We have now added two sections, one on experimental evolution and one on GWAS + perturbation. We hope this is able to capture the type of answers both approaches can offer, and to distinguish this from a classic GWAS where there is no perturbation. In the experimental evolution section, we have made more emphasis on how by designing the starting population, specific questions can be addressed, which contrasts with classic GWAS.

I agree that experiments using populations of genetically polymorphic model organisms in different controlled environments have a lot to add to our knowledge of the genetic architecture of quantitative traits. Larger samples always give more power, whole genome DNA sequencing gives a complete picture of genetic variation, especially if long reads are employed, and automated phenotyping of multiple traits on the same individuals will resolve pleiotropy. The addition of RNA sequencing and other 'omics analyses of dissected tissues and at different times of development would also provide the opportunity to understand how variants are associated with the phenotypes via various network levels (i.e., systems genetics). However, there are experimental designs other than parallel selection experiments that can achieve the same goal, and the disadvantages of the parallel selection paradigm are not addressed.

In conclusion, the authors fail to outline what "mechanisms underlying phenotypic heredity" need to be discovered by the new field of "experimental polygenic genetics". This is a Perspectives article, so the authors have leeway in expressing their opinions. However, rather than clearly outlining challenges and potential solutions, the paper becomes a plea for funding their own research programs.

> We have edited the text to convey more clearly 1) which type of geneticists we are trying to address with this perspective, 2) what are the limitations with the current one-gene-one-trait approach, 3) what are the challenges in embracing polygenicity, and 4) what experimental approaches can be taken (going beyond experimental evolution). We hope that this presents a clearer and more balanced view.

Reviewer #3 Comments:

Andersson et al. make an argument that the field of genetics needs to more fully embrace the complexity of most phenotypes and that reaching this goal will also require a shift to new experiments. They provide the historical context for this argument and highlight several areas of genetics where major gaps in knowledge remain. I wholeheartedly agree that an overly simplistic view of the genetic basis of phenotypes has been too prevalent in the field of genetics and that this perspective being entrenched in the field has limited the advancement of knowledge. Thus, I do find this perspective to be valuable and needed in the field. However, I also have some suggestions for the framing of these issues and some concerns in the current draft that I detail below.

1) Increased acknowledgement and discussion of those in the field who have embraced complexity in genetics

On line 356, the authors write, "The discovery of the highly polygenic nature of the generation of organismic phenotypes has come as a shock even to many specialists." Not just in this one section, but throughout the manuscript, I felt the tone was that pretty much all geneticists shared the simplistic view of genetics they describe. I think it is important to point out that 1) in fact many people (probably nearly all scientists who describe themselves as quantitative geneticists) were not at all surprised, 2) that arguments emphasizing the importance of recognizing the likely highly complex nature of the genetic basis of most phenotypes have been made before several times (a couple examples include: Rockman 2012 Evolution; Travisano & Shaw 2013 Evolution), and 3) that there is a rich, long history of work that has embraced polygenicity & complexity from the

very beginning that has produced a lot of valuable knowledge about the emergent properties of genetics (if not about specific gene mechanisms).

>We copy here the response to similar comments from other reviewers:

"We agree that our focus on describing how we got to the one-gene-one-trait view could be understood as if nothing else had been done in terms of quantitative genetics. We have now re-structure the first section of the manuscript that is now called 'How did we get here? The birth of genetics, and the establishment of the dominant view' to include a section on the theoretical and experimental developments that provide the data for polygenic architectures. This section is now called 'Genetics without genes' to serve the narrative where we try to explain how we got to the current view."

We have also edited the wording across the whole manuscript to make it clear that we are trying to talk to researchers that still use 'classical genetics' approaches and not to the whole community of geneticists, which as you point out includes many people and fields that have embraced complex trait genetics for a long time.

And, we have also added explicit mention to people that before us have also tried to call for the community to embrace the G-P complexity. Thanks for the suggested references.

I understand the decision to emphasize and spend the most time detailing the viewpoint that is being argued against here. However, I think it is equally important to point out that there has been an alternative viewpoint and well-developed theory that predicted high complexity. While the authors do acknowledge Fisher's infinitesimal model and the later omnigenic model, I did find their coverage of the history of the field to be uneven with a strong emphasis on the simplistic view. I was also surprised that the modern synthesis was not discussed more. The authors describe the divide that existed before the modern synthesis but don't fully explain how the first time the field realized: "whether heredity is based on particulate elements or continuous variation has been resolved: it is both!", as they state in the conclusion, was during the modern synthesis in the 1930's and 40's (Provine's book, *The Origin of Theoretical Population Genetics* is a nice discussion of this history). It may be useful to point out more explicitly that attempts to bring these perspectives together have been made repeatedly throughout this history but there has been continual resistance to a full acknowledgement and embrace of complexity.

> Please see the previous response.

2) Clarification of what exactly the field needs to "revisit."

The authors use a couple different phrases that they argue the field needs to revisit, including "the mechanistic basis of phenotypic inheritance" and "the mechanisms of heredity". I think if you say those phrases to different people, you will get different answers about what they think they mean. I also think that for many, the phrase will imply that the argument is suggesting that the fundamental misunderstanding that is being discussed has to do with how individual alleles are inherited and/or some other aspect of the mechanisms of gene action. However, I don't think this is actually what the authors think needs changing. I would argue that there is not anything about mechanism that is not completely compatible with a model of most traits being highly polygenic and context dependent. Isn't the argument more about an increased openness to ideas like: traits being determined by many, many variants and the effects of a given variant typically being context dependent? From my perspective, these ideas don't challenge the fundamental mechanism of how variants are inherited but are instead different hypotheses about the characteristics of the variants underlying a given phenotype. I think the perspective would be more effective if it stated very clearly what it means (and does not mean) by these phrases. Similarly, the authors use "Mendelian" as a short-hand for the simplistic perspective they are arguing against but elsewhere note that they are not arguing against the idea that "discrete Mendelian variants are at the basis of heredity." So, while I think I understand the perspective

they mean when they say, "Mendelian", I don't think it is the best term to encompass the viewpoint given that we use this as a way to accurately describe how individual variants are inherited. I would encourage the authors to spend some time fully defining the perspective they are arguing against (e.g. "The dominant perspective in the field of molecular genetics has implicitly assumed that 1) 2).... 3)....") and give this perspective a name that is not also the name of a legitimate process not in question so that you can refer to it easily throughout but also be sure that the reader has the same understanding of what is meant.

>We thank the reviewer for this comment, we now appreciate that the wording we chose could create confusion regarding what exactly we want the community to think about. We have changed 'Mendelian genetics' and similar wording for 'classical genetics', or 'mechanisms of heredity' by 'genotype-phenotype relationships', and included a simple definition of what we mean with classical genetics to guide the rest of the manuscript.

3) More careful consideration of what needs citation and which papers to cite.

I am sympathetic to the scope of the ideas being presented here and the sheer number of references that could be cited for a given idea. However, I do think there are fewer citations than needed and that in some cases fundamental references are missing. I'd encourage the authors to go back through and assess where they have statements that need a citation but have none. This is a general comment but a couple of examples of what I mean are below.

On line 155, they introduce the idea of a GxE, a concept with a very long history with many papers and books discussing its definition, potential role, and research findings. They choose to cite a single, relatively recent review paper here. I felt this choice left out many important citations and also fails to show the volume of work in this area that already exists.

> We have now included more references where we thought they were missing, but keeping in mind that this is a perspective, not a review, and we therefore have decided to focus on more recent citations that represent better the current state of the field, but that can serve as starting points for going deep, if a reader wishes.

Both the history section and the section on new paradigms especially seemed to have many statements without a citation that I think need some. For example, "Artificial selection experiments have been performed for decades, hence the general understanding of how they work is well established.", and, "And they need a good number of replicates to allow statistical analysis of allele frequency changes. The latter example in particular should cite the several simulation studies that have demonstrated this fact.

>We have added two references to simulations, as also requested by reviewer #1.

4) A more complete discussion of what the limitations and the potential results are for the new paradigms they suggest.

The authors make an argument for a fairly specific experimental program as a way to move the field forward: experimental evolution studies. However, they are a bit vague on exactly how they imagine the resulting datasets will better inform our understanding of which gene variants contribute to which phenotypes and what the expected effects of these variants are in different contexts. In particular, there is not discussion of potential limitations that will remain or what might be challenging in taking this approach. I am very much in favor of and in support of an experimental evolution approach, but it is also true that there are real challenges to understanding which variants contribute to adaptation using this approach and that many of the realities of polygenicity will also present limitations when using this approach. I think the authors should acknowledge this.

> We agree - but the goal is to raise attention to new experimental paradigms in the first place. We discuss challenges and unsolved issues throughout the manuscript. This is actually the very motivation for calling towards expanding this research program. Only the future can show how these challenges will eventually be solved.

December 22, 2025
RE: GENETICS-2025-308797

Dear Dr. Tautz:

I am pleased to accept your Perspectives titled "Beyond Mendel: a call to revisit the genotype-phenotype map through new experimental paradigms" for publication in GENETICS, pending minor revision.

Please submit your revision along with a response to the reviewers' concerns and suggestions, which can be viewed at the bottom of this email. Basically one minor concern about being more explicit in the discussion of epistasis (and its relationship to background effects).

I expect this can be done within 30 days.

Additionally, please ensure that your revision is formatted for GENETICS: <https://academic.oup.com/genetics/pages/general-instructions>.

Follow this link to submit the revised manuscript: Link Not Available

Thank you for submitting your research to Genetics.

Sincerely,

Anthony Long
Senior Editor
GENETICS

Approved by:
Howard Lipshitz
Editor in Chief
GENETICS

Reviewer comments:

Reviewer #2 :

I commend the authors for taking seriously the comments of the reviewers of the first submission. This revision is significantly improved and makes a valuable contribution to the literature. I have one minor comment regarding the sections on suppressor alleles and genetic background-specific effects. These two phenomena are epistatic interactions; however, the term epistasis is not used. Perhaps because it was coined by Bateman, a Mendelist? Or because human geneticists deny the existence of epistasis because it contributes little epistatic variance? However, uncovering epistatic interactions are arguably a key to truly understanding polygenicity. The sections as written are excellent; I would like to see them tied together and more emphasis placed on the importance of epistasis. The authors do mention gene-gene interactions, but we have a perfectly good technical term that defines these.

Reviewer #3 :

Andersson et al. have made substantial revisions leading to a much improved perspective. I found the description of the challenges and history of the field much clearer and more accurate of where things currently stand. In addition, I appreciated the change in structure to emphasize other potential experimental approaches and to more clearly delineate the main points of the perspective. I do still think that there will be some big challenges associated with embracing complexity while still aiming to decipher genetic mechanisms but the perspective now more clearly discusses some of these limitations and makes it clear that these are a potential starting point. I believe this discussion will be of high value to the genetics community.

Reviewer comments:

Reviewer #2 :

I commend the authors for taking seriously the comments of the reviewers of the first submission. This revision is significantly improved and makes a valuable contribution to the literature. I have one minor comment regarding the sections on suppressor alleles and genetic background-specific effects. These two phenomena are epistatic interactions; however, the term epistasis is not used. Perhaps because it was coined by Bateman, a Mendelist? Or because human geneticists deny the existence of epistasis because it contributes little epistatic variance? However, uncovering epistatic interactions are arguably a key to truly understanding polygenicity. The sections as written are excellent; I would like to see them tied together and more emphasis placed on the importance of epistasis. The authors do mention gene-gene interactions, but we have a perfectly good technical term that defines these.

Response: We agree that the term epistasis was somewhat lost from the text, but the concept was of course covered. We have now explicitly added sentences and references to epistasis.

Reviewer #3 :

Andersson et al. have made substantial revisions leading to a much improved perspective. I found the description of the challenges and history of the field much clearer and more accurate of where things currently stand. In addition, I appreciated the change in structure to emphasize other potential experimental approaches and to more clearly delineate the main points of the perspective. I do still think that there will be some big challenges associated with embracing complexity while still aiming to decipher genetic mechanisms but the perspective now more clearly discusses some of these limitations and makes it clear that these are a potential starting point. I believe this discussion will be of high value to the genetics community.

Response: Thanks for this assessment. We agree that our perspective can only be a start for a much more complex discussion on this topic.

January 17, 2026

RE: GENETICS-2025-308797R1

Prof. Diethard Tautz
Max-Planck-Institut für Evolutionsbiologie
Evolutionary Genetics
August-Thienemann-Strasse 2
Ploen, N/A D-24306
Germany

Dear Dr. Tautz:

Congratulations, your Perspectives titled "Beyond Mendel: a call to revisit the genotype-phenotype map through new experimental paradigms" is accepted for publication in GENETICS! Many thanks for contributing to GENETICS.

To Proceed to Publication:

1. Format your article according to GENETICS style: <https://academic.oup.com/genetics/pages/author-guidelines>
2. Ensure that you comply with data and community resource citation guidelines: <https://academic.oup.com/genetics/pages/author-guidelines#section-5-9-2>
3. Upload your final files at <https://genetics.msubmit.net>
4. Add oupsupport@scipris.com and genetics.oup@novatechset.com (or the domains @scipris.com and @novatechset.com) to your email program's "safe senders" list. You will be contacted by both at various points during the production process.

Notes:

- We invite you to submit an original color figure related to your paper for consideration as cover art. Please email your submission to the editorial office or upload it with your final files. You can submit a small-sized image for evaluation, and if selected, the final image must be a TIFF file 2513px wide by 3263px high (8.375 by 10.875 inches; resolution of 600ppi). Please avoid graphs and small type.

- After files are sent to Oxford University Press we use SciPris to manage article licensing and payment. If you do not have a SciPris account, you will receive an email from no-reply@scipris.com to sign up to use Oxford University Press' author portal. After logging in, follow the online instructions to sign your licence. It is important that you select the Standard License to Publish so that the GSA will be billed for the page charges (Open Access is not covered by the GSA).

If you have any questions or encounter any problems while uploading your accepted manuscript files, please email the editorial office at sourcefiles@thegsajournals.org.

Sincerely,

Anthony Long
Senior Editor
GENETICS

Approved by:
Howard Lipshitz
Editor in Chief
GENETICS